# Involvement of Long Non-Coding RNAs (lncRNAs) in Tumor Angiogenesis

**DOI:** 10.3390/ncrna6040042

**Published:** 2020-09-25

**Authors:** Julia Teppan, Dominik A. Barth, Felix Prinz, Katharina Jonas, Martin Pichler, Christiane Klec

**Affiliations:** 1Research Unit of Non-Coding RNAs and Genome Editing in Cancer, Division of Clinical Oncology, Department of Internal Medicine, Comprehensive Cancer Center Graz, Medical University of Graz, 8036 Graz, Austria; julia.teppan@medunigraz.at (J.T.); dominik.barth@medunigraz.at (D.A.B.); felix.prinz@medunigraz.at (F.P.); katharina.jonas@medunigraz.at (K.J.); christiane.klec@medunigraz.at (C.K.); 2Department of Experimental Therapeutics, the University of Texas MD Anderson Cancer Center, Houston, TX 77030, USA

**Keywords:** long non-coding RNA (lncRNA), angiogenesis, tumor, cancer

## Abstract

Long non-coding RNAs (lncRNAs) are defined as non-protein coding transcripts with a minimal length of 200 nucleotides. They are involved in various biological processes such as cell differentiation, apoptosis, as well as in pathophysiological processes. Numerous studies considered that frequently deregulated lncRNAs contribute to all hallmarks of cancer including metastasis, drug resistance, and angiogenesis. Angiogenesis, the formation of new blood vessels, is crucial for a tumor to receive sufficient amounts of nutrients and oxygen and therefore, to grow and exceed in its size over the diameter of 2 mm. In this review, the regulatory mechanisms of lncRNAs are described, which influence tumor angiogenesis by directly or indirectly regulating oncogenic pathways, interacting with other transcripts such as microRNAs (miRNAs) or modulating the tumor microenvironment. Further, angiogenic lncRNAs occurring in several cancer types such as liver, gastrointestinal cancer, or brain tumors are summarized. Growing evidence on the influence of lncRNAs on tumor angiogenesis verified these transcripts as potential predictive or diagnostic biomarkers or therapeutic targets of anti-angiogenesis treatment. However, there are many unsolved questions left which are pointed out in this review, hence driving comprehensive research in this area is necessary to enable an effective use of lncRNAs as either therapeutic molecules or diagnostic targets in cancer.

## 1. Introduction

While 80% of the human genome is transcribed into RNA, less than 2% of the genome is translated into proteins [1]. For quite a long time, the non-translated region of the genome was wrongly viewed as “junk DNA”. Transcripts of this area represent the group of non-coding RNAs as they are never translated into proteins [2]. In the last decades, science focused on the relatively new field of non-coding RNAs, discovering multiple subgroups. In general, they can be classified into small non-coding RNAs, including families such as microRNA (miRNA), endogenous small interfering RNA (siRNA), piwi interacting RNA (piRNA), and long non-coding RNA (lncRNA) [3].

LncRNAs are non-coding transcripts with a minimal length of 200 nucleotides and are transcribed by the enzyme RNA polymerase II or RNA polymerase III. Similar to mRNAs, most lncRNAs are 5′ capped, spliced, and polyadenylated after transcription [4,5]. Compared to mRNAs, lncRNAs contain fewer but longer exons and their expression levels across different tissues are commonly lower [5,6]. Interestingly, recent studies unexpectedly discovered peptides encoded from non-coding RNAs such as lncRNAs, which were considered to lack translational potential. Functional peptides translated from lncRNAs underline the significance of their encoding potential [7]. Matsumoto et al. published the lncRNA LINC00961 translated polypeptide, named small regulatory peptide of amino acid response (SPAR). SPAR inhibits the activation of mammalian target of rapamycin complex 1 (mTORC1), which is part of the mTOR pathway and controls cellular growth [8]. LincRNA00908 encodes a polypeptide, which indirectly influences angiogenesis by binding the transcription factor Signal Transducer And Activator Of Transcription 3 (STAT3), and hence inhibits its phosphorylation, which decreases Vascular Endothelial Growth Factor (VEGF) expression [9]. LncRNAs are further divided according to their genomic localization into: (a) sense lncRNAs, which are located on the same strand as the protein-coding gene, (b) antisense lncRNAs, which are located on the antisense strand of the protein-coding gene, (c) bidirectional lncRNAs, which are transcribed in the opposite direction from the promotor, (d) intronic lncRNAs, which originate from introns, and (e) intergenic lncRNAs (lincRNA), which are located between two genes [10] (Figure 1).

In the last few years, lncRNAs received growing attention and were characterized as “key players” in various hallmarks of cancers due to their regulatory function in physiological processes, like cell differentiation and apoptosis, as well as their involvement in pathophysiological processes [11,12,13]. LncRNAs have a broad range of interaction partners inside and outside the nucleus, and therefore play a role in diverse cellular mechanisms. One the one hand, lncRNAs can bind to nucleic acids such as DNA, mRNAs, or miRNAs [14]. On the other hand, proteins and peptides are known to be potential interaction partners forming complexes with lncRNAs. Furthermore, lncRNAs contain structural elements such as loops, that enable binding of small-weight molecules. Kazimierczyk et al. [14] published a comprehensive review about the lncRNA interactome focusing on the lncRNA interactions in more detail. This large interactome of lncRNAs (Figure 2) involves them in a variety of regulatory processes such as chromatin remodeling, DNA methylation, or histone modification. Further, lncRNAs are involved in transcriptional, post-transcriptional, as well as epigenetic regulation [5,14]. Numerous studies have confirmed an aberrant expression of lncRNAs in various cancer types such as liver, gastrointestinal, or brain cancers, and discovered their crucial role in tumorigenesis. Due to their influence on gene expression, lncRNAs impact carcinogenesis by acting either as tumor-suppressive or oncogenic lncRNAs [15].

Angiogenesis is known as the process of growing new blood vessels, which is physiologically essential for evolving organs, wound healing, and pregnancy. In general, angiogenesis is distinguished in vasculogenesis, which describes forming new blood vessels during embryogenesis, and the classical angiogenesis including the prenatal growth and remodeling of the primary vascular network. In adults, angiogenesis is transiently activated for processes such as wound healing [16]. Moreover, this angiogenic pathway is adapted under pathological conditions in many diseases including cancer, in which it may be constantly activated because of the angiogenic switch [10,11].

In 1971, Folkman described tumor angiogenesis for the first time as a hallmark of cancer, which enables metastasis and tumor growth beyond a limited size [17,18]. To exceed a tumor size with a diameter of 2 mm, the formation of blood vessels is vital in order to sufficiently supply the tumor with nutrients and oxygen. The angiogenic switch is guided by signaling molecules such as the vascular endothelial growth factor (VEGF) which induces angiogenesis and neovascularization in cancer [19]. Beside such signaling factors, recent studies consider lncRNAs as possible drivers for tumor angiogenesis [4,20].

This review aims to provide an insight into the function of lncRNAs in tumor angiogenesis, emphasizing their regulatory mechanisms among different tumor types.

## 2. Tumor Angiogenesis

As described above, if tumors reach a diameter of 2 mm, cancer cells undergo the angiogenic switch recruiting new blood vessels to ensure an adequate supply of oxygen and nutrients. Newly constructed blood vessels enable the disposal of metabolic waste and facilitate metastatic spread, as they provide tumor cells a way into the blood flow [19,21].

Under hypoxia, the state of low oxygen, cancer cells prevent hypoxia inducible factors (HIFs) from degradation. HIF is a heterodimeric transcription factor consisting of a cytoplasmic α-subunit and a β-subunit [17]. Usually, HIF-α is degraded upon initial hydroxylation, which enables the recognition by the von Hippel-Lindau (VHL) tumor suppressor protein, thereby targeting it for the degradation by the proteasome [18]. However, under hypoxic conditions, which are frequently encountered in cancer, HIF-α is translocated into the nucleus, where it forms an active heterodimer with the β-subunit. In turn, the heterodimer binds to hypoxia response elements, further leading to aerobic glycolysis known as the Warburg effect or the activation of angiogenesis [22,23]. Thereby, lncRNAs increase the expression of proangiogenic factors such as VEGF. The VEGF family and its receptors (VEGFR) play a major role in tumor angiogenesis by activating multiple signaling pathways such as endothelial proliferation, vascular permeabilization, or mobilization of progenitor cells, to name a few examples [24]. Beside VEGF, a variety of proangiogenic regulators might be released, including diverse growth factors: fibroblast growth factor (FGF), epidermal growth factor (EGF), transforming growth factor (TGF), and angiopoietin 1 and 2 (Ang-1 and Ang-2). Another key process in tumor angiogenesis is the activation of oncogenic pathways like the signal transducer and activator of transcription 3 (STAT3) signaling pathway, and the target of rapamycin (mTOR) pathway including Nuclear Factor Kappa B (NF-κB) and Wnt/β-catenin pathways, which further promote the release of proangiogenic factors [25]. Furthermore, cancer cells recruit immune cells such as macrophages, mast cells, and neutrophils to infiltrate the tumor, contributing to a complex tumor microenvironment (TME). The so-called tumor-associated macrophages (TAM) support tumor angiogenesis by releasing matrix metalloproteinases (MMPs). This group of growth factors is comprised of 23 members, which degrade extracellular matrix (ECM) as well as membrane components between the tumor and the blood vessel [26,27]. Therefore, they enable the release of matrix-bound VEGF, binding to their endothelial receptor and stimulating endothelial cells to activate the mitogen-activated protein kinase (MAPK) signaling pathway, resulting in enhanced cell proliferation [24,26]. In contrast to proangiogenic factors, tissue inhibitors of metalloproteinases (TIMPs), interleukins, interferons, and thrombospondin-1 (TSP-1) also influence the complex mechanisms of tumor angiogenesis by acting as antiangiogenic regulators [28]. Finally, the released proangiogenic regulators recruit three types of stem cells from the bone marrow, leading to tube formations. While endothelial progenitor cells (EPCs) construct the lumen of new angiogenic sprouts, hematopoietic stem cells act as proangiogenic stimulators, tissue remodeling, as well as endothelial survival factors. Further, mesenchymal stem cells turn into pericytes stabilizing the newly formed blood vessels for the tumor [21].

The complex cascade of the pathological angiogenic process clearly demonstrates the importance of the TME, which includes both cells that are present under non-cancerous conditions, as well as cells that are recruited by the tumor such as the mesenchymal stem cells [29]. Beside recruited stem cells from the bone marrow, cancer stem cells are also involved in tumor angiogenesis. They form a critical subpopulation of tumor cells with self-renewable capacity that drives nearly every step of tumorigenesis, including angiogenesis [30]. Cancer cells evolved a contact-free way to stimulate angiogenesis by releasing exosomes to remodel the TME, which can further influence tumor angiogenesis. Several studies have shown that tumor-derived exosomal lncRNAs can be taken up by neighboring cells to regulate angiogenesis [31].

## 3. Mechanisms of lncRNAs Regulating Tumor Angiogenesis

Tumor angiogenesis is driven by a deranged equilibrium of factors that stimulate or oppose angiogenesis by binding to surface receptors of vascular endothelial cells [17] (Figure 3). Recent studies reported that lncRNAs might serve as key regulators in tumor angiogenesis by regulating tumor-associated cells, influencing oncogenic pathways directly and indirectly, or by binding other RNA transcripts [4,20]. This section will provide an overview of the different possibilities of lncRNAs to regulate tumor angiogenesis and give some important examples, respectively.

### 3.1. LncRNAs Regulate Angiogenesis through Activating Pathways or by Binding Proteins in Tumor Cells

Altered expression levels of lncRNAs associated with oxygen-deficient conditions have been reported in many cancer types [32,33,34]. Due to the current state of knowledge, most hypoxia-related lncRNAs are involved in the HIF pathway. LncRNAs can either regulate the HIF pathway upstream by acting as a promoter of HIF or function as a direct downstream target of HIF [20]. Yao et al. [34] published lncRNA-HIF2PUT as a direct upstream promoter inducing HIF-2α expression in colorectal cancer and therefore increasing angiogenesis. The activation of the transcription factor HIF-2 upregulates the expression of its downstream target lncRNA Nuclear Paraspeckle Assembly Transcript 1 (NEAT1) in breast cancer, resulting in an increased cell proliferation [35]. In addition to the possibility of lncRNAs acting as an up- or down-stream target, either positive or negative feedback loops promote or suppress angiogenesis, owing to the strong co-expression of some lncRNAs and their corresponding mRNAs [20]. For instance, due to their co-expression, lncRNA hypoxia-inducible factor-1 alpha subunit antisense RNA 2 (HIF-1A-AS2) binds to HIF-1α-mRNA, forming a negative feedback loop by preventing angiogenesis in non-tumor tissue. However, the exact pathway is not yet uncovered [4]. Different oncogenic pathways may be activated to increase angiogenesis by inducing growth factors such as the major contributor VEGF. Some lncRNAs activate or stabilize STAT3 and therefore indirectly increase VEGF expression [9,36,37,38]. For instance, lncRNA plasmacytoma variant translocation 1 (PVT1) activates STAT3, which leads to an upregulation of VEGF in gastric cancer. A positive feedback loop further increases the angiogenic capacity of PVT1 [36]. Similarly, lncRNAs were found to activate oncogenic Phosphatidylinositol-4,5-Bisphosphate 3-Kinase/AKT Serine-Threonine Kinase/mammalian target of rapamycin (PI3K/Akt/mTOR) or Wnt/β-catenin signaling, resulting in the upregulation of VEGF. Li et al. [39] published the proangiogenic role of lncRNA OR3A4 by indirectly upregulating VEGF through the activation of the Akt/mTOR pathway in hepatocellular cancer. Several studies indicate an overexpression of lncRNA growth arrest-specific 5 (GAS5) reducing angiogenesis via inhibiting the Wnt/β-catenin signaling axis in cancer cells [40]. Other lncRNAs regulate angiogenesis by modulating the transcription factor NF-κB, such as LINC01410 in gastric cancer [41]. Tabruyn et al. [42] recently reviewed the role of NF-κB in tumor angiogenesis by highlighting its controversial proangiogenic function in tumor cells, while an activation of NF-κB inhibits angiogenesis in endothelial cells.

However, lncRNAs are not limited to act as decoys or bind and guide transcription factors to promoter regions of genes involved in angiogenic pathways, as described in some examples above. Furthermore, lncRNAs can also be associated with proteins to inhibit their secretion. For instance, the lncRNA associated with microvascular invasion in hepatocellular carcinoma (MVIH) interacts with the antiangiogenic factor phosphoglycerate kinase 1 (PGK1) to reduce its secretion in hepatic cancer [43]. A protein–lncRNA interaction can also affect the stability of the protein. For example, linc00665 binds the Y-box binding protein 1 (YB-1 protein), thus preventing it from degradation, and subsequently increasing angiogenesis in lung cancer [44]. Furthermore, lncRNAs are able to regulate enzymatic activity by binding to proteins, as illustrated by the interaction between lncRNAs and proteins of the Matrix Metallopeptidase (MMP) family. As these enzymes are either upstream or downstream targets in different signaling pathways including HIF, VEGF, and transforming growth factor beta (TGF-β), it affects tumor angiogenesis [27]. Metastasis-associated lung adenocarcinoma transcript 1 (MALAT-1) is a conserved lncRNA, whose expression is increased under hypoxia and which is associated with tumor angiogenesis. It was shown that this lncRNA promotes MMP7 expression via an indirect activation through the Wnt/β-catenin signaling pathway, thus facilitating angiogenesis in colorectal cancer [20].

### 3.2. Interaction of lncRNAs with miRNAs or mRNAs

Another group of non-coding RNAs, which is involved in many cellular processes, is the family of miRNAs. They negatively regulate gene expression by binding to the three prime untranslated region (3′UTR) of mRNAs, leading to the degradation, destabilization, or translational repression of the respective mRNA targets [45]. In 2005, Yang and colleagues [46] first indicated the role of miRNAs in a study of angiogenesis conducted in mouse models. Nowadays, a variety of either proangiogenic miRNAs or antiangiogenic miRNAs are known in different cancer types [47].

A further possibility of lncRNAs to indirectly influence tumor angiogenesis is the interaction with these miRNAs which are modulating their activity, and vice versa: (1) LncRNAs can act as molecular sponges for miRNAs, resulting in a repressed function of the miRNA. The mechanism of competitive endogenous RNA (ceRNA) has been widely investigated in tumor angiogenesis. (2) Furthermore, it was shown that binding of miRNAs to lncRNAs might lead to their degradation [4,20,25]. (3) Another possible way to affect each other is the competition of some lncRNAs and miRNAs for the same mRNA. (4) Other lncRNAs are able to generate miRNAs. These interactions of non-coding RNAs clearly state the complexity of the interplay between non-coding RNAs and tumor angiogenesis as they are involved in many biological pathways [4].

To give some examples for discovered non-coding RNA interactions, MALAT-1 was found to sponge miRNAs. The decrease of miR-200 and miR-145 leads to an upregulation of the protein SRY-Box transcription factor 2 (Sox2), and hence increases stem cell properties [4]. Further, in lymphoma and glioblastoma cells, MALAT-1 is targeted by miR-9 for decay inside the nucleus. Besides the interaction with MALAT-1, miR-9 also affects angiogenesis by targeting VEGF [48]. The crosstalk between two noncoding RNAs, that both have an impact on tumor angiogenesis, is an example for the complex regulation of tumor angiogenesis modulated by non-coding RNAs. Another example is lincRNA-p21, which is in general activated by p53 and binds to let-7 miRNA, thereby inhibiting its angiogenic functions [49]. On the contrary, results by Yoon et al. [50] indicate that an overexpression of let-7 miRNAs in cervical cancer leads to a degradation of lincRNA-p21. Many studies have investigated the ceRNA functions of lncRNAs acting as molecular sponges for miRNAs in different cancer types. Both nuclear and cytoplasmic lncRNAs can modulate angiogenesis by inhibiting miRNA activity [51]. The lncRNA taurine upregulated 1 (TUG1) can either act as a tumor suppressor or as oncogenic lncRNA [6]. In endometrial carcinoma, TUG1 enhances the expression of Vascular Endothelial Growth Factor A (VEGFA) by directly binding miR-34a-5p and miR-299 [52]. The extended regulation of angiogenesis by lncRNAs through miRNA interaction offers a relatively new research field, also with possibilities for therapeutic discoveries [4].

Beside proteins and miRNAs, in some cases, lncRNAs can bind to mRNAs and influence the angiogenesis this way. An example is the lncRNA MVIH, which interacts with the mRNA of ribosomal protein S24 isoform C, found in colorectal cancer. Herein, it promotes angiogenesis through an increased stability of both transcripts [53].

### 3.3. LncRNAs Affecting Neighboring Cells of the Tumor

Beside influencing angiogenesis through mechanisms within tumor cells, lncRNAs can affect the function of adjacent cancer cells, non-cancerous cells in the TME, or surrounding endothelial cells, to promote angiogenesis.

Cancer stem cells (CSCs) are a small subgroup of cancer cells that provide self-renewal capacity similar to stem and progenitor cells. These cells play a regulatory role in every step of tumor progression, from tumor initiation to metastasis formation. Recent studies showed the impact of lncRNAs on angiogenesis by regulating CSCs [30,54]. As an example, Jiao et al. [55] provided data indicating that MALAT-1 induces epithelial–mesenchymal transition in pancreatic cancer cells, thereby gaining stem cell-like properties. Furthermore, MALAT-1 knockdown leads to a decreased expression of CSC markers [55]. Advanced studies of this group suggest the proangiogenic effect of MALAT-1 by promoting stem cell-like phenotypes with the binding of miR-200c and miR-145, thereby upregulating the self-renewal factor Sox2 [56]. The crosstalk between CSCs and the TME mediate signaling pathways such as TGF-β, MAPK, or VEGF, further influencing tumor angiogenesis [30,54].

LncRNAs can also remodel the microenvironment of the tumor. As described above, macrophages infiltrate the tumor and constitute the major component of the TME. On the one hand, lncRNAs can recruit macrophages to infiltrate the tumor by activating a downstream signaling of cytokines, such as the lncRNA CamK-A [25]. On the other hand, lncRNAs are involved in the polarizations of macrophages. For instance, lncRNA-MMP2 was found to influence TME remodeling as it correlates with the level of M2 polarization. Further, silencing lncRNA-MMP2 in macrophages inhibits M2-induced angiogenesis [57].

LncRNAs can be delivered to endothelial cells via tumor-derived exosomes to promote angiogenesis in endothelial cells. Exosomes are membrane-bound extracellular vesicles with a diameter of 30–100 nm, which are released by living cells. Recent studies have demonstrated this contact-free way to modulate angiogenesis [31]. As an example, glioma cells release exosomes loaded with lncRNA-POU Class 3 Homeobox 3 (POU3F3), which are in turn internalized with endothelial cells. As a consequence, lncRNA-POU3F3 expression levels increase within these cells and angiogenesis is promoted in vitro and in vivo [58].

## 4. LncRNAs Regulating Tumor Angiogenesis in Different Cancer Types

Several lncRNAs were reported to increase or decrease angiogenesis in various cancer types, modulating this hallmark of cancer by different intracellular processes (Table 1). For instance, lncRNA TUG1 promotes angiogenesis in cervical cancer by an exosomal transport of the lncRNA to the receptor human umbilical vein endothelial cells (HUVECs), thus enhancing their angiogenic capacity [59]. In comparison, the same lncRNA accelerates angiogenesis by upregulating VEGF expression through miRNA binding, namely miR-299 in glioblastoma [60] and miR-34-5p in liver cancer [52]. Likewise, angiogenic lncRNA MALAT-1 and highly upregulated in liver cancer (HULC) induce hepatic angiogenesis by sponging miRNAs as well. Additionally, less common lncRNAs such as Colorectal Neoplasia Differentially Expressed (CRNDE) [61], Opa interacting protein 5-antisense RNA 1 (OIP5-AS1) [62], and LINC00488 [63] are also known to induce liver cancer angiogenesis by functioning as ceRNA. In the following section, angiogenic lncRNAs are summarized by their regulatory mechanisms occurring in different tumor types.

### 4.1. Hepatic Cancer

As mentioned in the previous section, the lncRNA OR3A4 is upregulated in hepatocellular carcinoma (HCC), which is the most frequent liver cancer type. OR3A4 accelerates angiogenesis via activation of the Akt/mTOR pathway and thereby further increases the secretion of VEGF [39]. On the contrary, tumor-suppressive lncRNA PTENP1 expression is decreased in HCC cells. PTENP1 overexpression suppresses the PI3K/Akt axis and inhibits HIF, leading to suppressed angiogenesis in vitro [65]. The lncRNA BZRAP1-AS1 binds to the promotor of thrombospondin (THB1), an endogenous antiangiogenic factor, and recruits DNA methyltransferase 3b, thus inducing the hypermethylation of the THB1 promoter and leading to the inhibition of THB1 transcription. The THB1 depletion further increases VEGF expression, thereby promoting angiogenesis in HCC [64]. By activating ERK/HIF-1α signaling, the overexpressed lncRNA, named ubiquitin conjugated enzyme E2C pseudogene 3 (UBE2CP3), promotes the VEGF secretion from HCC cells into the TME [66]. The study of Yuan and colleagues [43] determined that lncRNA MVIH binds to the antiangiogenic factor PGK1. This interaction leads to an inactivation of PGK1, which accelerates angiogenesis in HCC.

As the name indicates, HULC (highly upregulated in liver cancer) was first characterized to be highly upregulated in HCC. The interaction between HULC and miR-107 results in the upregulation of E2F Transcription Factor 1 (E2F1), which subsequently results in the activation of the sphingosine kinase 1 (SPHK1) promoter. Consequently, SPHK1 generates sphingosine 1 phosphate, which was found to accelerate angiogenesis in liver fibrosis, HCC, and breast cancer [67]. In hepatoblastoma (HB), another type of liver cancer, the highly expressed lncRNAs CRNDE and TUG1 were shown to bind miR-203 and miR-34-5p, respectively. By acting as a ceRNA, these lncRNAs prevent the suppressive miRNA function on the target gene VEGF [52,61]. Therefore, lncRNA TUG1 might be responsible for the hypervascularity, which is characteristic for hepatoblastoma [52]. Similar results were found in HCC, where lncRNAs MALAT-1 and OIP5-AS1 induce angiogenesis because of an upregulated VEGF expression by sponging miR-140 and miR-3163, respectively [62,69]. Another example of lncRNAs promoting tumor angiogenesis in liver cancer by their ceRNA function is LINC00488. This lncRNA sponges miR-330-5p, thereby upregulating the protein talin 1 in relation to angiogenesis in HCC [63]. Beside LINC00488, Dong et al. [68] suggested lncRNA LEF1-AS1 to promote angiogenesis in HCC. This hypothesis is based on knockdown experiments showing significantly reduced tube formations as compared to control conditions. Mechanistically, the authors hypothesized that the binding of LEF1-AS1 and miR-136-5p may further upregulate its target WNK1. WNK1 itself was shown to be associated with an increased level of MMP2, MMP9, and VEGF expression [68].

Generally, CSCs are known to influence the TME by releasing signaling extracellular organelles, the exosomes. Conigliaro et al. [71] described the ability of cluster of differentiation 90 (CD90+) liver CSCs to promote an angiogenic phenotype of endothelial cells by releasing exosomes containing lncRNA H19. This study highlights lncRNA H19 as an important player of exosome-mediated VEGF increase in HCC [71]. The group of Ye [70] investigated the impact of lncRNA cox-2 expression in macrophage polarization, that is commonly uncontrolled in HCC. A transient siRNA-mediated knockdown of lncRNA cox-2 reduced the polarization of M1-macrophages and accelerated M2 macrophage polarization, leading to increased HCC angiogenesis [70].

### 4.2. Gastrointestinal Cancer

The lncRNAs PVT1, TPT1-AS1, and FLANC are highly expressed in gastrointestinal cancers and have been shown to increase tumor angiogenesis by protein interactions inside the nucleus, which further lead to an upregulation of VEGF [36,37,80]. The group of Zhang [80] recently reported that the binding of lncRNA TPT1-AS1 to the nuclear factor 90 (NF90) increases the mRNA stability of VEGF and therefore facilitates angiogenesis in colorectal cancer (CRC). VEGF expression is increased by STAT3 activation through lincRNAs PVT1 and FLANC and hence, angiogenesis rises in gastric cancer (GC) and CRC, respectively [36,37]. Beside the correlation of PVT1 and VEGF expression by preventing STAT3 from degradation, the oncogenic function of this lncRNA is enhanced by a positive feedback loop with STAT3, as it promotes PVT1 transcription [36]. Further, lncRNA SUMO1P3 is highly expressed in colon cancer cells and tissue, while a knockdown decreased tube formations, VEGF expression, and angiogenesis in vivo [79]. LncRNAs GAS5 [40], OR3A4 [33], and LINC01314 [75] are found at low expression levels in gastrointestinal cancer. An overexpression of these lncRNAs leads to a decline of angiogenesis via repressing the Wnt/β-catenin pathway [33,40,75]. Tang et al. [75] investigated that an upregulation of LINC01314 decreased kallikrein 4 expression and further inhibited the Wnt/β-catenin pathway, as well as the expression of VEGF in GC. The knockdown of lincRNA AK001058 also decreased angiogenic cytokine expression of VEGF and Ang-2 in CRC through a hypermethylation of the metallopeptidase ADAM Metallopeptidase With Thrombospondin Type 1 Motif 12 (ADAMTS12) promoter [72]. Recently, the group of Xu [74] described the angiogenic role of linc00858 in colon cancer. This lncRNA facilitates angiogenesis by an upregulation of the transcript factor Hepatocyte Nuclear Factor 4 Alpha (HNF4α) to sustain a repression of WNK Lysine Deficient Protein Kinase 2 (WNK2), a kinase that functions as growth suppressor [74]. LncRNA HNF1A-AS1 also promotes colon cancer angiogenesis by an indirect activation of the MAPK pathway [73]. In GC, the activity of lncRNA neuroblastoma-associated transcript 1 (NBAT1) is suppressed by the interaction with the transcription factor SOX9, forming a negative feedback loop. Yan and colleagues [78] indicated that the antiangiogenic function of NBAT1 as an overexpression reduced tube formations in vitro. Besides its role in HCC, MVIH is suggested to modulate angiogenesis in CRC indirectly as this lncRNA is stabilized by its binding to the mRNA of ribosomal protein S24 isoform c. Hence, it causes a repression of PGK1 and promotes angiogenesis [53]. Oncogenic lncRNA MALAT-1 enhances tumor angiogenesis either in CRC or GC [76,77]. In vitro experiments propose MALAT-1 as an angiogenic regulator in GC by modulating the vascular endothelial (VE)-cadherin/β-catenin complex, which regulates vascular permeability [77].

Sun et al. [76] showed that in CRC, MALAT-1 can bind and suppress miR-126-5p, whose target is VEGF. In response to the nuclear transcription factor SP1, the upregulated lncRNA ZNFX1 antisense RNA1 (ZFAS1) acts as a sponge for miR-105 in CRC. The induced repression of miR-105 due to this direct binding consequently promotes VEGF expression [82]. Zhang and colleagues [41] investigated the ceRNA function of LINC01410 for miR-532, which results in an upregulation of neutrophil cytosolic factor 2 and the activation of the NF-κB pathway. Activated NF-κB sustains an upregulation of LINC01410, forming a positive feedback loop in GC [41]. Further, lncRNA-HNF1A-AS1 increases tube formation and facilitates GC angiogenesis in vivo. Mechanistically, lncRNA-HNF1A-AS1 sponges miR-30b-3p, thereby activating the PI3K/Akt signaling pathway [81].

Wang et al. [83] hypothesize that lncRNA-APC1 would inhibit angiogenesis by reducing the exosome production in CRC cells. Previously, it has been reported that tumor-derived exosomes promote angiogenesis without the necessity of common angiogenic factors [83].

In addition to the above-mentioned lncRNAs, the upregulated lncRNA OR3A4 and downregulated lncRNA cancer susceptibility 2 (CASC2) were also found to influence GC angiogenesis. However, the exact mechanism is not yet known [33,84].

### 4.3. Brain Tumor

Recently, Wang et al. [87] described an increased expression of lncRNA protein disulfide isomerase family A member 3 pseudogene 1 (PDIA3P1) in glioblastoma cells, which is activated at the transcriptional level by the HIF heterodimer. PDIA3P1 can further sponge miR-124-3p, which is found to inhibit angiogenesis by targeting NRCP1 in glioblastoma [87,120]. Zhu et al. [85] reported the correlation between HULC and micro-vessel density, VEGF expression, and endothelial cell-specific molecule 1, which is suggested to modify angiogenic capacity of HULC through PI3K/Akt/mTOR signaling in gliomas. An overexpression of lncRNA SLC26A4-AS1 was associated with antiangiogenic effects in glioma [88]. Yu et al. [88] hypothesized a lncRNA-dependent recruitment of NF-κB1 to the promotor of neuronal pentraxin 1, causing an upregulation and hence, decreasing angiogenesis. The lncRNA PAX-interacting protein 1-antisense RNA1 (PAXIP1-AS1) enhances angiogenesis in glioma by recruiting a transcription factor to the KIF14 promotor, which is a prognostic marker in glioma patients as it has been associated with glioma aggressiveness in prior studies [86,142].

In brain tumors, lncRNAs TUG1, LINC01116, X-inactive specific transcript (XIST), SNHG15, and RNA SBF2 antisense RNA 1 (SBF2-AS1) are suggested to induce angiogenesis by their ceRNA function on different miRNAs [60,91,93,94,95]. In glioblastoma cells, the highly expressed lncRNA TUG1 favors angiogenesis by inhibiting miR-299, which has a binding site for VEGF [60]. The data of Ye et al. [91] showed that a LINC01116 knockdown suppresses glioma angiogenesis in vivo. They suggest that LINC01116 may regulate posttranscriptional VEGF by binding miR-31-5p. Similar, XIST sponges miR-485 in human brain microvascular endothelial cells (HBMEC). Hence, miR-485 binding enables SOX7 to further increase VEGF expression level [95]. Additionally, Cheng et al. [96] identified a correlation of lncRNA XIST and miR-429 in glioma cells affecting angiogenesis. The knockdown of XIST reduces glioma angiogenesis in vitro and in vivo [96]. Likewise, the knockdown of lncRNA SNHG15 reduced the expression levels of proangiogenic Cdc42 and VEGF because miR-153 is no longer sponged and subsequently, tumor angiogenesis in glioma cells is suppressed [94]. In the work of Hai et al. [93], lncRNA SBF2-AS1 was identified to promote glioblastoma angiogenesis by binding to miR-338, thereby accelerating the expression and secretion of EGFL7. On the contrary, the lncRNA micro-chromosome maintenance protein 3-associated protein (MCM3AP-AS1) is targeted by miR-211 and coupled in the RNA-induced silencing complex (RISC). As a result, when the miRNA target Krüppel-like factor 5 is activated, glioblastoma angiogenesis is increased via the downstream regulation of angiogenic factor AGGF1 [92]. Furthermore, lncRNA H19 facilitates glioblastoma cell angiogenesis in vitro and in vivo [89,90]. Angiogenic function of H19 in glioma-induced endothelial cells is enabled by miR-29a suppression to inhibit the proangiogenic vasohibin 2 [90].

In the work of Dai et al. [97], the lncRNA antisense transcript of hypoxia factor-1α (AHIF) is revealed as a regulator of exosomal segregation of VEGF in glioblastoma. Several studies have shown the potential of glioma cells to enhance angiogenesis by lncRNA containing exosomes. This contact-free way to transport proangiogenic transcripts such as POU3F3, HOX Transcript Antisense Intergenic RNA (HOTAIR), or Colon cancer associated transcript 2 (CCAT2) to endothelial cells leads to an increased expression level of VEGF and thereby promotes angiogenesis [58,98,99].

### 4.4. Reproductive System Cancer

The group of Iden [102] found that lncRNA PVT1 is upregulated in cervical cancer cell lines in response to hypoxia, which is associated with poor patient prognosis. Furthermore, an overexpression of lncRNA cancer susceptibility candidate 2 (CASC2) in cervical cancer inhibits angiogenesis in vitro via the MAPK pathway [100]. The knockdown of the oncogenic lncRNAs TUG1 and LINC00284 diminished angiogenesis in ovarian cancer [101,103]. TUG1 may inhibit angiogenesis in vitro by suppressing TGF-β signaling pathway via the downregulation of the angiogenic factor leucine-rich-α-2-glycoprotein-1. Additionally, TUG1 induces other angiogenic growth factors such as VEGF [103]. LINC00284 favors angiogenesis by recruiting NF-κB to the mesoderm-specific transcript (MEST) promotor to repress the transcription, consequently increasing the expression of proangiogenic proteins such as MMPs or VEGF [101].

The upregulated lncRNA differentiation antagonizing non-protein coding RNA (DANCR) was first reported to increase angiogenesis in ovarian cancer. Knockdown experiments exhibit an increased miR-145 expression, which is a direct target of DANCR and finally leads to a downregulation of VEGF [104]. Recently, highly expressed lncRNA SCAMP1 was also found to accelerate angiogenesis in ovarian cancer as it disabled miR-137 function by binding. MiR-137 acts as a negative regulator of the C-X-C motif chemokine, which participates in gastric cancer progression [109]. The loss of maternally expressed gene 3 (MEG3), which encodes a tumor-suppressive lncRNA, is associated with tumorigenesis in many cancer types as it mediates various cellular factors and pathways, such as p53, proangiogenic genes, or miRNAs [4,105]. Ye and colleagues [106] investigated a regulatory axis of lncRNA MEG3, miR-421, and the growth factor platelet-derived growth factor receptor α (PDGFRA) by binding to each other. An overexpression of the lncRNA RBMS3-AS3, which is usually downregulated in prostate cancer, demonstrated the ceRNA function of the lncRNA on miR-4534. By inhibiting the function of miR-4534, expression of the antiangiogenic vasohibin-1 rises. Consequently, expression levels of the proangiogenic factors MMP2, MMP9, and VEGF increase, thereby promoting angiogenesis [108]. Fang’s work showed that transient silencing of the prostate cancer-specific lncRNAs PCAT3 and PCAT9 decreases tumor angiogenesis. The sequences of both lncRNAs include miR-203 binding sites, which enables an increased expression of the SNAI2 protein [107].

In ovarian cancer, a crosstalk between epithelial ovarian cancer cells (EOCs) and endothelial cells may facilitate angiogenesis as EOCs release exosomes including MALAT-1. Exosomal MALAT-1 enhances angiogenesis by dysregulating angiogenic genes like VEGF in HUVEC cell models and in vivo [111]. Similarly, Lei and Mou proposed lncRNA TUG1 to be transported via exosomes from cervical cancer cells to HUVEC cells, increasing their angiogenic capacity. Furthermore, silenced TUG1 expression is associated with a decreased expression level of several proangiogenic genes such as VEGF, MMP9, and TGF-β [59].

Additionally, Ding et al. [110] showed that the downregulation of lncRNA CCDST influences angiogenesis in cervical cancer. Investigations of the underlying mechanisms indicated that the lncRNA binds the proangiogenic DHX9 and targets it for degradation. However, in human papillomavirus (HPV)-infected cervical cancer, the virus-encoded proteins lead to a decrease of CCDST expression and consequently, to the increase of DHX9 abundance to induce malignant behaviors [110].

### 4.5. Lung Cancer

The lncRNA LINC00667 is strongly expressed in non-small cell lung cancer and stabilizes the angiogenic growth factor VEGF [112]. Wang et al. [38] pointed out that the lncRNA TNK2-AS1 binds STAT3 to stabilize it and increases the expression of VEGF. Furthermore, STAT3 promotes lncRNA TNK-AS1 transcription, building a positive feedback loop [38]. In lung adenocarcinoma, lncRNAs LOC100132354 and linc00665 influence angiogenesis by regulating VEGF expression [44,114]. Mechanistically, linc00665 stabilizes YB-1 protein from degradation, hence activating VEGF expression, leading to increased angiogenesis in vitro and in vivo [44]. In contrast to the detected downregulation of MEG3 in lung cancer, Li et al. [105] published data detecting an enhanced expression of MEG3 in lung adenocarcinoma, promoting VEGF-mediated angiogenesis by activating Akt signaling. This hypothesis proposes MEG3 as a cell-specific angiogenesis promotor, as Lui and colleagues [115] detected a significant downregulation of MEG3 in lung adenocarcinoma. The expression of the upregulated lincRNA-p21 correlates with the expression of angiogenic genes such as MMP2 presenting its angiogenic function [113]. As an overexpression of the lncRNA, MVIH is associated with tumor angiogenesis and MVIH expression is upregulated in lung cancer, further influencing MMP2 and MMP9, it may affect lung cancer tumor angiogenesis [116].

The upregulated lncRNAs PVT1 and FBXL19-AS1 both accelerate angiogenesis by acting as a ceRNA on miR-29 and miR-431-5p, respectively [118,122]. Apart from brain tumors, lncRNA MCM3AP-AS1 is also upregulated in lung cancer binding to miR-340-5p. Therefore, the downstream target KPNA4 is activated and promotes angiogenesis [121]. Quin et al. [117] propose an antiangiogenic function of lncRNA F630028O10Rik (F63) by sponging miR-223-3p, which causes a downregulation of VEGF expression. The downregulated lncRNA GAS5 can affect angiogenesis negatively by binding to miR-29-3p together with phosphate and tensin homolog (PTEN). Subsequently, this inhibits the phosphorylation of PI3K/Akt and thereby decreases VEGF expression [119,120].

Furthermore, the GAS5 expression level in tumor-derived exosomes correlates with tube formations as exosomes of lung cancer cells having low GAS5 expression promoting tube formation and vice versa [119].

Besides the previously mentioned lncRNAs, oncogenic lncRNA HOXA11-AS is highly expressed in non-small cell lung cancer, regulating angiogenesis positively. However, the exact molecular mechanism is still unclear [123].

### 4.6. Breast Cancer

In response to hypoxia, lncRNA EFNA3 is induced in breast cancer (BC), leading to an accumulation of Ephrin-A3, both facilitating metastatic spread [124]. Niu et al. [128] discovered lncRNA RAB11B-AS1 as a novel target of HIF2 leading to an upregulation of angiogenic factors such as VEGF. As mentioned before, the lncRNA HIF-1A-AS2 seems to act as a counter to tumor angiogenesis as its co-expression with HIF1α leads to a negative feedback loop. As lncRNA HIF-1A-AS2 binds to the corresponding mRNA, resulting in its degradation. However, the exact mechanism has not yet been uncovered [4]. The expression of lncRNA MEG3 is lost in different tumors including BC, which leads to an increased expression of proangiogenic genes. MEG3 overexpression suppresses the PI3K/Akt signaling pathway further, leading to a decreased expression of VEGF [126]. Similarly, Luo et al. [127] recently showed that the overexpression of lncRNA NF-κB Interacting lncRNA (NKILA) inhibits angiogenesis by reducing IL-6 secretion via modulation of the NF-κB signaling pathway. The lncRNA MALAT-1 is upregulated in human BC tissue and its expression correlates with VEGF, suggesting MALAT-1 as another angiogenic lncRNA in BC [125]. Recently, Wang et al. [9] investigated that lincRNA00908 encodes a small regulatory peptide of STAT3 (ASPRS), which is downregulated in triple-negative BC. ASRPS directly binds to STAT3, therefore inhibiting its phosphorylation, which decreases VEGF expression [9].

LINC00968 acts as a ceRNA by binding the miRNA hsa-miR-423-5p in BC. Therefore, a decreased amount of LINC00968 induces an increased expression of its binding partner. Hsa-miR-423-5p further downregulates PROX1 and inactivates its function to repress MMP14, thereby promoting angiogenesis [129].

Zhou and colleagues [130] described lncRNA-Hh, which is transcribed by EMT cells, to obtain CSC-like properties [20,130].

### 4.7. Other Cancer Types

The commonly upregulated lncRNA MALAT-1 upregulates proangiogenic factors such as VEGF and FGF2 in osteosarcoma by activating the mTOR/HIF-1α signaling pathway as well as via a positive feedback loop of HIF-1α and the lncRNA [133]. In nasopharyngeal carcinoma, lncRNA HOTAIR is highly expressed and enhances angiogenesis in vitro and in vivo. Mechanistically, the lncRNA regulates angiogenesis either directly through binding to the VEGF promotor or indirectly via increasing VEGF and Ang-2 expression by upregulating glucose-regulated protein 78 expression [131]. In clear cell renal cell carcinoma, a knockdown of lncRNA POU3F3-adjacent non-coding transcript 1 (PANTR1) leads to a reduction of angiogenic parameters and less tube formations [134].

Recently, lncRNA SNHG6 and DANCR were identified to accelerate angiogenesis in cholangiocarcinoma by competitively binding miRNAs, thus leading to an increased expression of transcriptional factors [135,137]. In detail, SNHG6 binds miR-101-3p to prevent the inhibition of the transcription of E2F Transcription Factor 8 (E2F8) [137], whereas the DANCR–miR-345-5p interaction upregulates Twist1. In osteosarcoma, proangiogenic lncRNA TUG1 functions via silencing miR-143-5p, increasing the miRNA target HIF-1α [139]. Zhao et al. [136] identified linc00511 promoting angiogenesis in pancreatic ductal adenocarcinoma by competitively binding to hsa-miR-29b-3p. The induced repression of this miRNA finally promotes the VEGF expression and upregulates its expression [136]. Chi et al. [138] proposed lncRNA RP11-79H23.3 functioning as a ceRNA and further regulating tumorigenesis of bladder cancer by modulating the PTEN/PI3K/Akt pathway. They demonstrated that an overexpression reduces angiogenesis in vivo, whereas a downregulation increases the amount of formed micro-vessels within the tumor [138].

As mentioned above, current studies provided an insight on the correlation between tumor angiogenesis and CSCs. Jiao et al. [56] suggested that MALAT-1 promotes pancreatic angiogenesis by enhancing stem cell-like properties. Huang et al. [132] investigated the proangiogenic function of MALAT-1 in thyroid cancer by regulating the FGF2 secretion of TAMs. Chondrosarcoma cells transport lncRNA RAMP2-AS1 in exosomes and thereby promote angiogenesis in HUVECs through sponging miR-2355-5p and hence, upregulate the expression of VEGFRs [140].

The commonly dysregulated lncRNA H19 is upregulated in bladder cancer cells and tissue and promotes angiogenesis in vivo [141].

## 5. Therapeutic Potential

The studies of lncRNAs regulating tumor angiogenesis have given strong evidence that they might be a promising target in cancer therapy, as a deregulation of these non-coding RNAs can lead to pathological changes.

Nowadays, different possibilities to regulate lncRNA expression by RNA interference in viral vectors or plasmids are known. For instance, cancer therapy using a plasmid containing H19 gene regulator sequence and diphteria toxin A (DTA) was tested in clinical trials in bladder, pancreatic, and ovarian cancer [143]. For silencing lncRNA expression, siRNAs are a widely used tool. For instance, 1,2-dioleoyl-sn-glycero-3-phosphatidylcholine nanoparticles loaded with FLANC targeting siRNA reduces angiogenesis and metastasis, demonstrating its potential as a therapeutic target in the future [37].

Another possibility to depress tumor angiogenesis offers a functional modulation such as altering the physiological interaction between lncRNA and miRNA. The option of functional regulation can be combined with gene therapy. For instance, Wang and colleagues [144] investigated an adenoviral vector, which includes lincRNA-p21 and a miRNA responsive element (MRE) of miR-451. Due to the MRE, the vector is delivered into CSC, preventing the activation of the Wnt/β-catenin pathway and finally eliminating CSCs in CRC [144]. In prior sections, the possibility of exosomal transport of non-coding RNAs between cells and its impact on angiogenesis was described. Cheng et al. [145] published a detailed review of the different options utilizing exosomes as potential therapeutic targets in glioma cancer. On the one hand, targeting exosomal transport would be a potential anticancer strategy. On the other hand, exosomes could be used as a therapeutic drug delivery system. Beside therapeutic options, exosomes can be used as predictive and diagnostic biomarkers [145].

Further, some drugs were found to regulate the expression levels of lncRNAs. Ye et al. [106] demonstrated anisomycin as a potential anticancer drug by suppressing angiogenesis, invasion, and proliferation by regulating the lncRNA MEG3 in ovarian cancer. As mentioned before, MEG3 sponges miR-421, further regulating the growth factor PDGFRA and activating the Notch pathway to induce tumor growth and angiogenesis. This ceRNA function of MEG3 can be inhibited by anisomycin [106].

Beside the possibility to act as a therapeutic target, circulating RNAs in plasma or serum build a non-invasive possibility for diagnostic application. Therefore, lncRNAs can also function as diagnostic biomarkers, although they might be stable as fragments in human liquids. In prostate cancer, lncRNA MALAT-1 can be used to distinguish patients from healthy controls [146]. Tong et al. [147] described plasmatic lncRNA POU3F3 as a diagnostic biomarker for esophageal squamous cell carcinoma patients. In combination with a current biomarker, the serum squamous cell carcinoma antigen, a more efficient diagnostic rate can be reached compared to the efficiency of RNA or antigen alone [147]. Furthermore, lncRNAs can function as key markers and specific lncRNA expression patterns even enable a differentiation between subtypes, for example in renal cell carcinoma [148].

Due to their specificity, lncRNAs may become useful biomarkers and outstanding therapy targets in cancer treatment. However, at present, the state of knowledge of lncRNAs and their role in diseases is incomplete. A considerable amount of research is necessary to enable an effective use of lncRNAs as either therapeutic molecules or diagnostic targets.

## 6. Conclusions

Studying lncRNAs has become an important research field as they are involved in basic cellular processes. Recent studies accentuated their critical role in different diseases, such as cancer. There are numerous examples of deregulated lncRNAs in various types of tumors, thereby contributing to different hallmarks of cancer, such as angiogenesis. In this review, we focused on the mechanisms of lncRNAs with which they regulate tumor-induced angiogenesis, including: (a) modulating oncogenic pathways directly or indirectly by binding pathway involved proteins, (b) interacting with RNA transcripts, or (c) modulating the TME. Key lncRNAs in angiogenesis such as MALAT-1 are deregulated in various cancer types, influencing angiogenesis by affecting different mechanisms. On the one hand, MALAT-1 is found to activate the Wnt/β-catenin pathway indirectly in response to hypoxia in CRC [20]. On the other hand, MALAT-1 also promotes CRC angiogenesis by its ceRNA function on miR-126-5p as the binding increases the expression of the miRNA target VEGF [76]. On top of that, tumor-derived MALAT-1 is transported in exosomes to HUVEC cells, enhancing angiogenesis by dysregulating proangiogenic genes like VEGF in vivo [111]. Angiogenesis-modulating lncRNAs can occur in different cancer types, influencing diverse mechanisms. Therefore, this article summarized angiogenic lncRNAs depending on their occurrence in different cancer types and further explained their regulatory mechanism concerning tumor angiogenesis. Due to their specificity, lncRNAs may become outstanding therapy targets in cancer treatment. Nowadays, studies establish different methods using gene therapies or functional modulations of lncRNAs to inhibit the process of angiogenesis. Besides the opportunity of being a therapeutic target, lncRNAs offer the potential of a diagnostic biomarker. To resume the prior given example, the angiogenic lncRNA MALAT-1 enables an effective diagnosis of prostate cancer through the detection of MALAT-1 fragments in human plasma [146]. Although lncRNAs show a huge potential due to the variety of regulatory mechanisms, there are still many unanswered questions. Understanding the complex process of lncRNAs regulating tumor angiogenesis may provide new antiangiogenic therapeutics as well as prognostic or diagnostic biomarkers in cancer.

## Figures and Tables

**Figure 1 ncrna-06-00042-f001:**
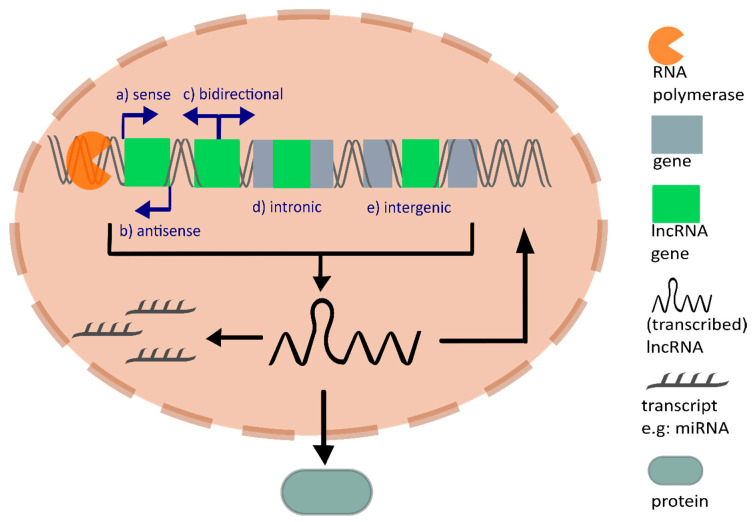
Schematic representation of long non-coding RNA (lncRNA) biogenesis and lncRNA function. According to their genomic localization, lncRNAs are divided into sense, anti-sense, bidirectional, intronic, or intergenic lncRNAs. They are transcribed by RNA polymerases and are often spliced, 5′ capped, and polyadenylated. Some lncRNAs are further translated into proteins, most of them function immediately after transcription on genome regulation or as binding partners for molecules inside (e.g., transcripts such as microRNAs (miRNAs)) or outside the nucleus (e.g., proteins).

**Figure 2 ncrna-06-00042-f002:**
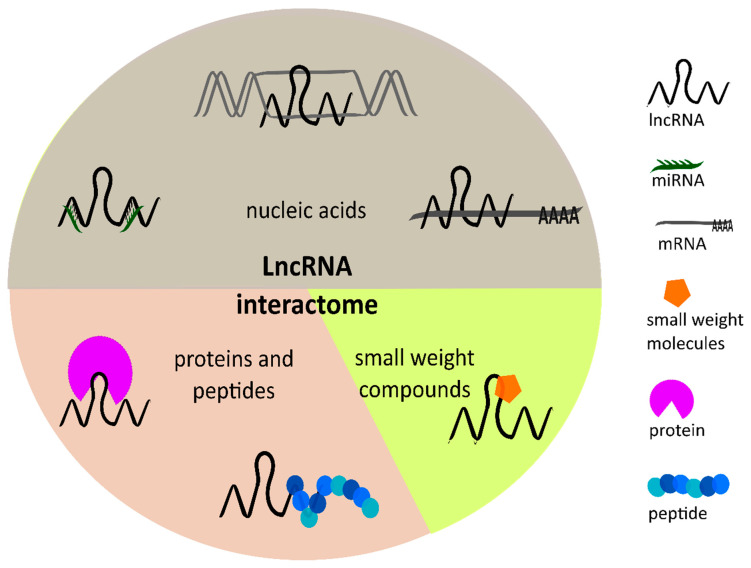
Overview of the lncRNA interactome. Possible interactions partners of lncRNAs are nucleic acids such as DNA, messenger RNA (mRNA), and miRNA, as well as proteins and peptides and small-weight compounds. Due to their broad variety of interaction partners, lncRNAs can further influence multiple cellular processes.

**Figure 3 ncrna-06-00042-f003:**
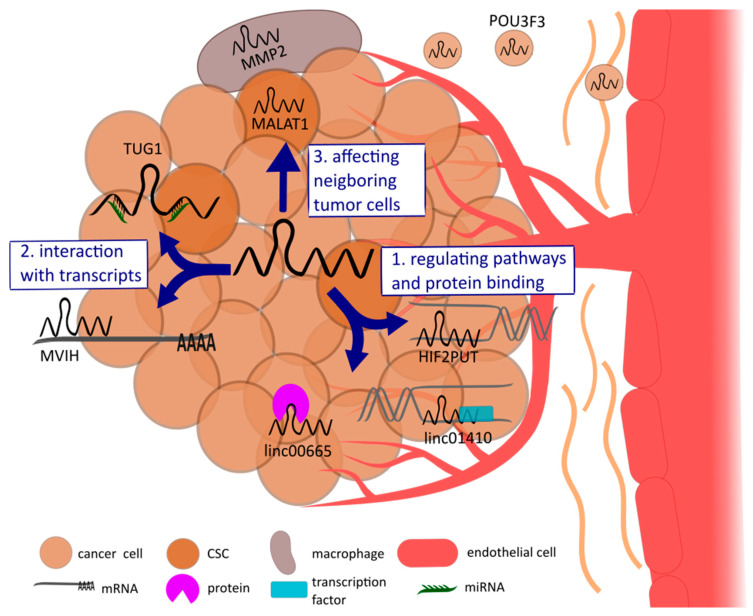
Schematic representation of the regulating mechanisms of lncRNAs influencing tumor angiogenesis: (1) protein binding and regulating pathways indirectly by recruiting transcription factors or directly affecting oncogenic pathways, (2) interacting with transcripts such as mRNA or miRNA, and (3) regulating neighboring cells, including the tumor microenvironment cells, e.g., cancer stem cells (CSC), macrophages, and endothelial cells. The latter mechanism is possible due to exosomal transport of lncRNAs. Exemplary lncRNAs for each shown mechanism are named in the figure.

**Table 1 ncrna-06-00042-t001:** Overview of long non-coding RNAs (lncRNAs) influencing tumor angiogenesis through different regulatory mechanisms in various cancer types.

LncRNA	Impact on Angiogenesis	Regulatory Mechanism	Reference
**1. Hepatic Cancer**			
BZRAP1-AS1	↑	Activating pathways or protein binding	[64]
MVIH	↑		[43]
OR3A4	↑		[39]
PTENP1	↓		[65]
UBE2CP3	↑		[66]
CRNDE	↑	Interacting with transcripts	[61]
HULC	↑		[67]
LEF1-AS1	↑		[68]
LINC00488	↑		[63]
MALAT-1	↑		[69]
OIP5-AS1	↑		[62]
TUG1	↑		[52]
cox-2	↓	Affecting neighboring tumor cells	[70]
H19	↑		[71]
**2. Gastrointestinal Cancer**			
AK001058	↑	Activating pathways or protein binding	[72]
FLANC	↑		[37]
GAS5	↓		[40]
HNF1A-AS1	↑		[73]
LINC00858	↑		[74]
LINC01314	↓		[75]
MALAT-1 ^1^	↑		[76,77]
MVIH	↑		[53]
NBAT1	↓		[78]
OR3A4	↓		[33]
PVT1	↑		[36]
SUMO1P3	↑		[79]
TPT1-AS1	↑		[80]
LINC01410	↑	Interacting with transcripts	[41]
lncRNA-HNF1A-AS1	↑		[81]
ZFAS1	↑		[82]
lncRNA-APC1	↓	Affecting neighboring tumor cells	[83]
CASC2	↓	unknown	[84]
OR3A4	↑		[33]
**3. Brain Tumor**			
HULC	↑	Activating pathways or protein binding	[85]
PAXIP1-AS1	↑		[86]
PDIA3P1	↓		[87]
SLC26A4-AS1	↓		[88]
H19	↑	Interacting with transcripts	[89,90]
LINC01116	↑		[91]
MCM3AP-AS1	↑		[92]
SBF2-AS1	↑		[93]
SNHG15	↑		[94]
TUG1	↑		[60]
XIST	↑		[95,96]
AHIF	↑	Affecting neighboring tumor cells	[97]
CCAT2	↑		[98]
HOTAIR	↑		[99]
POU3F3	↑		[58]
**4. Reproductive System Cancer**			
CASC2	↓	Activating pathways or protein binding	[100]
LINC00284	↑		[101]
PVT1	↑		[102]
TUG1 ^1^	↑		[59,103]
DANCR	↑	Interacting with transcripts	[104]
MEG3	↓		[4,105,106]
PCAT3	↑		[107]
PCAT9	↑		[107]
RBMS3-AS3	↓		[108]
SCAMP1	↑		[109]
CCDST	↓	Affecting neighboring tumor cells	[110]
MALAT-1	↑		[111]
**5. Lung Cancer**			
LINC00665	↑	Activating pathways or protein binding	[44]
LINC00667	↑		[112]
lincRNA-p21	↑		[113]
LOC100132354	↑		[114]
MEG3	↑		[4,105,115]
MVIH	↑		[116]
TNK2-AS1	↑		[38]
F63	↓	Interacting with transcripts	[117]
FBXL19-AS1	↑		[118]
GAS5 ^1^	↓		[119,120]
MCM3AP-AS1	↑		[121]
PVT1	↑		[122]
HOXA11-AS	↑	unknown	[123]
**6. Breast Cancer**			
EFNA3	↑	Activating pathways or protein binding	[124]
HIF-1A-AS2	↓		[4]
LINC00908	↑		[9]
MALAT-1	↑		[125]
MEG3	↓		[126]
NKILA	↓		[127]
RAB11B-AS1	↑		[128]
LINC00968	↑	Interacting with transcripts	[129]
lncRNA-Hh	↑	Affecting neighboring tumor cells	[130]
**7. Other Cancer Types**			
HOTAIR	↑	Activating pathways or protein binding	[131]
MALAT-1 ^1^	↑		[56,132,133]
PANTR1	↑		[134]
DANCR	↑	Interacting with transcripts	[135]
LINC00511	↑		[136]
SNHG6	↑		[137]
RP11-79H23.3	↓		[138]
TUG1	↑		[139]
RAMP2-AS1	↑	Affecting neighboring tumor cells	[140]
H19	↑		[141]

^1^ more than one regulatory mechanism is known in this tumor type. ↓ factor has anti-angiogenic effect; ↑ factor has pro-angiogenic effect. Abbreviations: AHIF: antisense transcript of hypoxia factor-1α; BZRAP1-AS1: benzodiazapine receptor associated protein 1 antisense RNA 1; CASC2: cancer susceptibility 2; CCAT2: Colon cancer associated transcript 2; CRNDE: Colorectal Neoplasia Differentially Expressed; DANCR: differentiation antagonizing non-protein coding RNA; EFNA3: Ephrin A3; F63: F630028O10Rik; GAS5: growth arrest-specific 5; HIF-1A-AS2: hypoxia-inducible factor-1 alpha subunit antisense; HOTAIR: HOX Transcript Antisense Intergenic RNA; HULC: highly upregulated in liver cancer; lncRNA-APC1: adenomatous polyposis coli; LEF1-AS1: lymphoid enhancer-binding factor 1; MALAT-1: Metastasis-associated lung adenocarcinoma transcript 1; MCM3AP-AS1: micro-chromosome maintenance protein 3-associated protein; MEG3: maternally expressed gene 3; MVIH: microvascular invasion in hepatocellular carcinoma; NBAT1: neuroblastoma-associated transcript 1; NKILA: Nuclear Factor Kappa B (NF-κB) Interacting lncRNA; OIP5-AS1: OPA-interacting protein 5 antisense transcript 1; OR3A4: Olfactory Receptor Family 3 Subfamily A Member 4; PANTR1: POU Class 3 Homeobox 3 (POU3F3)-adjacent non-coding transcript 1; PAXIP1-AS1: paired box (PAX)-interacting protein 1-antisense RNA1; PCAT3/9: Prostate Cancer Associated 3/9; PDIA3P1: protein disulfide isomerase family A member 3 pseudogene 1; PTENP1: phosphatase and tensin homolog pseudogene 1; PVT1: plasmacytoma variant translocation 1; SBF2-AS1: SBF2 antisense RNA 1; SCAMP1: secretory carrier membrane protein 1; SNHG6: small nucleolar RNA host gene 6; SNHG15: small nucleolar RNA host gene 15; SUMO1P3: Small ubiquitin-like modifier 1 pseudogene 3; TUG1: Taurine upregulated 1; UBE2CP3: ubiquitin conjugated enzyme E2C pseudogene 3; XIST: X-inactive specific transcript; ZFAS1: ZNFX1 antisense RNA1.

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
