# Peer review of "Involvement of Long Non-Coding RNAs (lncRNAs) in Tumor Angiogenesis"

_ncrna, 2020, doi:10.3390/ncrna6040042_

Round 1
Reviewer 1 Report
The authors give a very comprehensive overiew on current studies that link the non-coding transcriptome to the pathophysiology of angiogenesis in tumor development. In principle, the manuscript is publishable. However, I wish to mention that I didn´t enjoy reading it. The manuscript reads like a list of two-sentence synospses of roughly 150 papers. For a specialist in the field that might be satisfying, for someone like me who is not expert in the field I cannot take home much from that. The manuscript could be strengthened a lot if the authors could carve out the key concepts, the key findings, the key papers more clearly. What is cleary evidenced? What is currently controversial? What insight, what method, what achievement could bring the field forward? Is there any understanding deep enough to bring it into clinics? E.g., imagine you were an investor and wanted to invest into a clinical application targeting a noncoding RNA. What would be a promising approach? What research would be needed to de-risk clinical translation? Just some thoughts...
A minor point: according to the “author contribution table” and the sequence in the author list, the two last authors seem to have contributed similarly to the manuscript and may think about sharing the corresponding author position
Author Response
Reviewer 1
- The authors give a very comprehensive overiew on current studies that link the non-coding transcriptome to the pathophysiology of angiogenesis in tumor development. In principle, the manuscript is publishable. However, I wish to mention that I didn´t enjoy reading it. The manuscript reads like a list of two-sentence synospses of roughly 150 papers.
As we tried to provide a very comprehensive review without excluding any articles, lots of papers were included especially in the chapter 4 “LncRNAs regulating tumor angiogenesis in different cancer types”. To provide a better overview and readability we added a summarizing table (Table 1) at the beginning of this chapter including all lncRNAs and their mechanisms occurring in the different tumor types at a glance. This will also be helpful for the readers that might be interested in a special lncRNA/ cancer type.
- For a specialist in the field that might be satisfying, for someone like me who is not expert in the field I cannot take home much from that. The manuscript could be strengthened a lot if the authors could carve out the key concepts, the key findings, the key papers more clearly. Based on the reviewer´s comments the first chapter was partly rewritten to highlight the concepts of lncRNAs interacting with biomolecules which further enables them to influence biological process. For a better understanding, two figures were added explaining the biogenesis and the interactome of lncRNAs (Figures 1 and 2).
- What is cleary evidenced? What is currently controversial? What insight, what method, what achievement could bring the field forward? Is there any understanding deep enough to bring it into clinics? E.g., imagine you were an investor and wanted to invest into a clinical application targeting a noncoding RNA. What would be a promising approach? What research would be needed to de-risk clinical translation? Just some thoughts...
In chapter 5 “therapeutic potential” we tried to provide an insight on the current state of research and to highlight lncRNAs as possible biomarkers and potential therapeutic targets in cancer therapy. Indeed, this chapter only includes lncRNAs that were mentioned in the review before. However, adding new findings with elaborated context would make this review even longer. Therefore, we decided to keep the current structure of the present work.
- A minor point: according to the “author contribution table” and the sequence in the author list, the two last authors seem to have contributed similarly to the manuscript and may think about sharing the corresponding author position
We thank the reviewer for this note. All authors re-discussed the assignment of the authorship positions and came to the conclusion to leave it as it was.
Reviewer 2 Report
This review is well described and seems to be beneficial for researchers involved in this area. However, it seems to be difficult for researchers that are unfamiliar with this area, because there is only one schematic figure. The authors should add more schematic figures and tables including lncRNA list to this review for better understanding.
Author Response
Reviewer 2
- This review is well described and seems to be beneficial for researchers involved in this area. However, it seems to be difficult for researchers that are unfamiliar with this area, because there is only one schematic figure. The authors should add more schematic figures and tables including lncRNA list to this review for better understanding.
The manuscript now includes two more schematic graphs, providing an overview of the biogenesis and the interactome of lncRNAs i.e. Figure 1. “Schematic representation of long non-coding RNA (lncRNA) biogenesis and lncRNA function” (line 57-65) and Figure 2. “Overview of the lncRNA interactome”. Furthermore, the paragraph introducing lncRNAs, their biogenesis and functions was expanded (lines 41-57) and a table summarizing the impact of all described lncRNAs to tumor angiogenesis and the corresponding mechanisms i.e. Table 1.
Reviewer 3 Report
The manuscript by Teppan and co-workers is a comprehensive review on the role of long non-coding RNAs in tumor angiogenesis. Long non-coding RNAs are important new players in gene expression regulation, in cancer and beyond, and can act either directly or indirectly to affect the net outcome of myriads of regulatory non-coding RNAs. In this context I feel that the paragraph on the interaction of lncRNAs with miRNAs could be a little more expanded, since many researchers in the field would find it quite interesting. The manuscript is well written and includes several imprortant pieces of information regarding the involvement of specific lncRNAs in angiogenesis required for several cancer types.
The review merits publication in Non-codins RNAs, however prior to publication the authors might consider the following suggestions for additions that would be of more genral interest to the broader readership:
- A summarized table for each cancer type and lncRNA involved would be quite helpful for the readers
- An introductory paragraph on the biogenesis of lncRNAs is what most of similar reviews ommit. This information is important because it is still unclear for many cases whether the syncronization of transcription of specific mRNAs and lncRNAs is essential. In addtion, little is known on the possible maturation events of primary lncRNA transcirpts and the biochemistry behind it. Therefore, I feel that the addition of this information will be appreciated by the readers.
- The authors might consider to address on a small paragraph observations of lncRNAs that are translated. Although there are only few and in some cases contradictory reports, the authors could speculate on possible involvenent in carcinogenesis, since it has been reported that at least in some cases importnat signaling pathways (i.e. mTORC1) can be affected.
Author Response
Reviewer 3
- The manuscript by Teppan and co-workers is a comprehensive review on the role of long non-coding RNAs in tumor angiogenesis. Long non-coding RNAs are important new players in gene expression regulation, in cancer and beyond, and can act either directly or indirectly to affect the net outcome of myriads of regulatory non-coding RNAs In this context I feel that the paragraph on the interaction of lncRNAs with miRNAs could be a little more expanded, since many researchers in the field would find it quite interesting.
According to the reviewer’s suggestion more information on the interactions between lncRNAs and miRNAs was added and the corresponding paragraph was extended (lines 213 – 230).
- The manuscript is well written and includes several imprortant pieces of information regarding the involvement of specific lncRNAs in angiogenesis required for several cancer types.
The review merits publication in Non-codins RNAs, however prior to publication the authors might consider the following suggestions for additions that would be of more genral interest to the broader readership:
A summarized table for each cancer type and lncRNA involved would be quite helpful for the readers
We appreciate the suggestion of the reviewer. A summarizing table (Table 1) listing the described lncRNAs by their regulatory mechanisms occurring in different tumor types was added, highly improving the quality of the manuscript.
- An introductory paragraph on the biogenesis of lncRNAs is what most of similar reviews ommit. This information is important because it is still unclear for many cases whether the syncronization of transcription of specific mRNAs and lncRNAs is essential. In addtion, little is known on the possible maturation events of primary lncRNA transcirpts and the biochemistry behind it. Therefore, I feel that the addition of this information will be appreciated by the readers.
We took up the reviewer’s suggestion and added more information (lines 41-57) as well as a schematic figure of lncRNA biogenesis (Figure 1).
- The authors might consider to address on a small paragraph observations of lncRNAs that are translated. Although there are only few and in some cases contradictory reports, the authors could speculate on possible involvenent in carcinogenesis, since it has been reported that at least in some cases importnat signaling pathways (i.e. mTORC1) can be affected.
The reviewer makes an excellent point. Therefore, the manuscript now includes a small paragraph focusing on lncRNA encoded peptides. As examples, we included the lncRNA encoded peptide SPAR which modulates mTORC1. Further, we named the lincRNA00908 encodes a polypeptide named ASPRS, which regulates tumor angiogenesis indirectly, to mention their possible influence in tumorigenesis (lines 45-52).
Reviewer 4 Report
The authors review angiogenic lncRNAs in different types of cancer and explain their regulatory mechanisms. The subject is interesting and well explained. However, I do miss some tables and graphs that summarize the information at a glance. One proposal would be a graph with LncRNA interactions with other molecules and with cancer types.
Author Response
Reviewer 4
- The authors review angiogenic lncRNAs in different types of cancer and explain their regulatory mechanisms. The subject is interesting and well explained However, I do miss some tables and graphs that summarize the information at a glance. One proposal would be a graph with LncRNA interactions with other molecules and with cancer types.
The manuscript now includes a summarizing table (Table 1) listing the lncRNAs by their regulatory mechanisms occurring in different tumor types. According to the reviewer’s suggestion two schematic graphs were included, that give an overview of the biogenesis and the interactome of lncRNAs i.e. e. Figure 1. “Schematic representation of long non-coding RNA (lncRNA) biogenesis and lncRNA function” (line 57-65) and Figure 2. “Overview of the lncRNA interactome”. Furthermore, the review by Kazimierczyk et al. has been added and highlighted, which provides a comprehensive overview about the lncRNA interactome.
Round 2
Reviewer 4 Report
I agree with the authors' response and, in my opinion, the article is now ready for publication.